# Bridging Sustainable Development Goals and Land Administration: The Role of the ISO 19152 Land Administration Domain Model in SDG Indicator Formalization †

Mengying Chen [1],*, Peter Van Oosterom [1], Eftychia Kalogianni [1], Paula Dijkstra [2] and Christiaan Lemmen [3]

[1] Faculty of Architecture and the Built Environment, Delft University of Technology, 2628 BL Delft, The Netherlands; p.j.m.vanoosterom@tudelft.nl (P.V.O.); e.kalogianni@tudelft.nl (E.K.)

[2] Kadaster, The Netherlands Cadastre, Land Registry and Mapping Agency, 7311 KZ Apeldoorn, The Netherlands; paula.dijkstra@kadaster.nl

[3] Faculty of Geo-Information Science and Earth Observation (ITC), University of Twente, Hallenweg 8, 7522 NH Enschede, The Netherlands; c.h.j.lemmen@utwente.nl

* Correspondence: m.chen-21@student.tudelft.nl

† Presented at the 11th International FIG Workshop on the Land Administration Domain Model & 3D Land Administration, Gavle, Sweden, 11–13 October 2023.

**Abstract:** This study illustrates the linkages between the ISO's Land Administration Domain Model (LADM) and the UN's sustainable development goals (SDGs), highlighting the role of the LADM in promoting effective land administration suitable for efficient computation of land/water (space)-related SDG indicators. The main contribution of this study is the formalization of SDG indicators by using the ISO standard LADM. This paper proposes several SDG-indicator-related extensions to the multi-part LADM standard that is currently under revision. These extensions encompass the introduction of new procedures for calculating indicators, the integration of blueprints for external classes to fulfil additional information needs and the design of interface classes for presenting indicator values across specific countries and reporting years. In an innovative approach, this paper introduces the Four-Step Method—a powerful framework designed to formalize SDG indicators within the LADM framework. Detailed attention is devoted to specific indicators, including 1.4.2 (secure land rights), 5.a.1 (women's agricultural land rights), 14.5.1 (protected marine areas) and 11.5.2 (valuation as a basis for direct economic loss). In short, the Four-Step Method is pivotal in eliminating ambiguities, enhancing the efficiency of indicator computation and securing more accurate indicator values that more truly reflect the progress towards SDG realization. This approach is also expected to work with other (ISO) standards for other SDG indicators.

**Keywords:** LADM; SDGs; sustainable development; land registration; marine georegulation; valuation information

## 1. Introduction

The sustainable development goals (SDGs) provide a comprehensive framework for global action, encompassing critical areas such as poverty eradication, environmental sustainability, and social equity [1]. Among these goals, some are intricately tied to the realm of land, emphasizing the critical importance of effective land management and equitable land distribution for sustainable development [2]. In this context, land administration plays a pivotal role in ensuring the efficient management and just allocation of land resources [3].

The Land Administration Domain Model (LADM) is a prominent international standard (ISO 19152:2012) in the field of land administration, providing a comprehensive framework that defines conceptual models and standardized methodologies for the design and development of land administration systems [4]. The LADM serves as a crucial tool for nations seeking to enhance their land administration systems, thereby promoting

sustainable utilization of land resources and equitable distribution. The standard is currently under revision and in this paper, the new edition of the LADM is used [5]. Further, the revision of the LADM aims to support the computations of (relevant) SDG indicators as already indicated in the ISO stage-0 document for the future multi-part LADM [6,7]. How the indicator formalization can be realized is explained in this paper.

The objective of this paper is to explore and analyze how the SDG indicators can be linked with the LADM, aiming to support and advance the realization of SDGs through a formal methodology. Currently, SDG indicators are expressed in natural language, which can lead to ambiguity. The LADM provides a shared vocabulary (ontology) that helps to clarify these indicators, thereby reducing ambiguity. Specifically, this paper analyzes several specific indicators in more detail (Table 1): 1.4.2 (secure land rights) [8], 5.a.1 (women's agricultural land rights) [9], 14.5.1 (protected marine areas) [10] and 11.5.2 (valuation as a basis for direct economic loss) [11]. Methods and procedures are added to the existing LADM classes to perform the actual indicator calculations, along with blueprints for external classes with additional information needs and interface classes to display the resulting indicator values for a specific country in a specific reporting year. Ensuring the accuracy of indicator values is paramount, particularly in highlighting the true realization of SDGs. When data are available, the LADM helps to improve the efficiency of calculating SDG indicators and reduces ambiguity. In the absence of a land administration system, the LADM can assist in establishing a system and calculating related SDG indicators. This paper analyzes existing methods for calculating these indicators and explores the possibility of simplifying the calculation process by revising the LADM. The revised LADM can optimize data management and data updates, thereby supporting effective monitoring and realization of SDGs [6].

**Table 1.** Case study indicators and their corresponding abbreviations in this paper.

| Indicator No. | Indicator Full Name | Abbreviation in This Paper |
|---|---|---|
| 1.4.2 | "Proportion of total adult population with secure tenure rights to land, (a) with legally recognized documentation, and (b) who perceive their rights to land as secure, by sex and type of tenure." | 1.4.2 (secure land rights) |
| 5.a.1 | "(a) Proportion of total agricultural population with ownership or secure rights over agricultural land, by sex; and (b) share of women among owners or rights-bearers of agricultural land, by type of tenure." | 5.a.1 (women agricultural land rights) |
| 14.5.1 | "Coverage of protected areas in relation to marine areas." | 14.5.1 (protected marine areas) |
| 11.5.2 | "Direct economic loss attributed to disasters in relation to global gross domestic product (GDP)" | 11.5.2 (valuation as basis for direct economic loss) |

The rest of this paper is structured as follows. The remainder of the first section presents the necessary background information, and the concepts addressed in this paper are introduced: the basic concepts of the LADM, its ongoing revisions and a brief background of SDGs. The second section describes the methodology followed and material used in this paper. The third section gives the main results of this paper: the indicator development process and examples by selecting four representative indicators and applying the aforementioned processes. In the fourth section, the results will be discussed. Finally, the conclusions and proposals for future research are presented.

### 1.1. ISO 19152:2012 LADM Basic Concepts and Ongoing Revision

The Land Administration Domain Model (LADM) is an international standard which describes people-to-land relationships by providing a shared vocabulary (ontology) and a formal language (Unified Modeling Language, UML) [4], aiming to facilitate communication among various stakeholders, both within one country and internationally. While

the LADM is a generic model, it can be extended and customized for specific regions or countries, making it a versatile tool in the field of land administration (as presented in [12]).

Widely adopted by international organizations like the United Nations [13] and the World Bank, the LADM serves as a common language for different stakeholders such as land surveyors [14], land registrars [15] and land managers [16]. Currently, around ten countries around the world have implemented the LADM (or are working on this) as a part of their land administration systems, including Scotland, Indonesia, and Colombia, while more than fifteen countries have adopted the Social Tenure Domain Model (STDM) [17].

The scope of LADM Edition I is limited to the land tenure component of the land administration paradigm (see the grey circle in Figure 1), whereas LADM Edition II aims to extend the scope of Edition I to include land value, land use and land development (red circle in Figure 2 [6]).

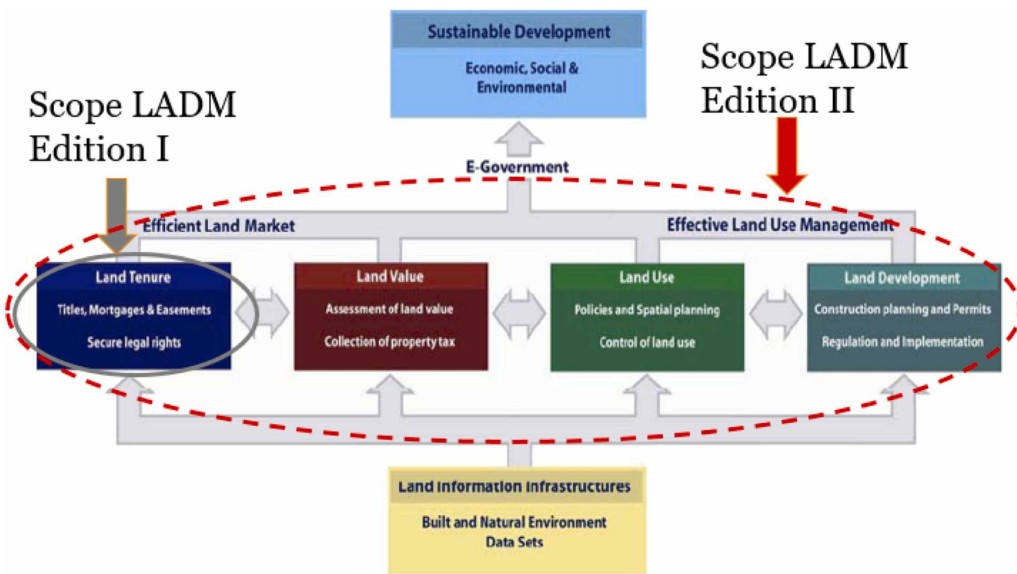

**Figure 1.** Land administration paradigm and LADM scope ([5]; adapted from [18]).

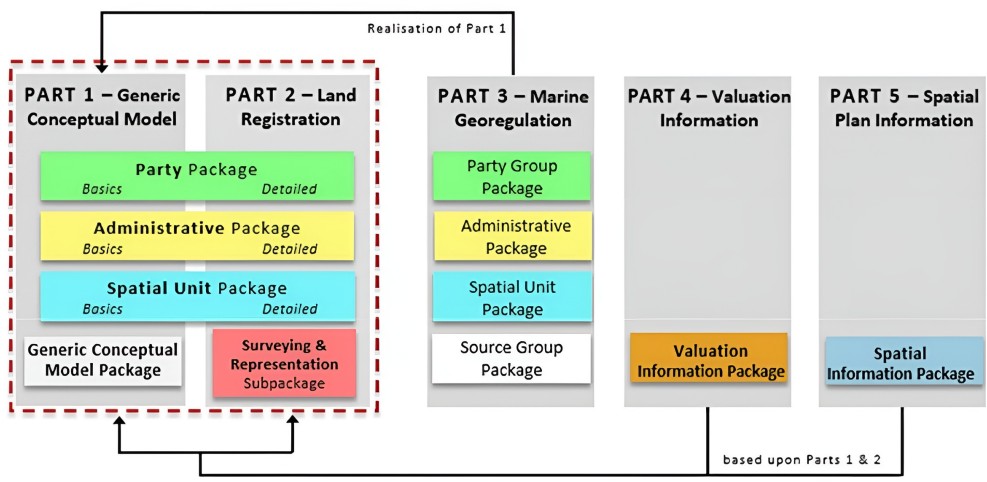

**Figure 2.** LADM Edition II Parts 1-5 ([19]).

Currently, the second edition of the LADM is under development, comprising six parts (Figure 2), each serving a distinct purpose within the realm of land administration. Part 1 serves as the overarching standard, providing a high-level outline for the entire framework. Part 2, building upon the foundation set by the LADM Edition I, expands 3D spatial profiles. Part 3 seeks to synchronize the delineation of RRRs with maritime concepts to bridge land

and marine domains. Part 4 is dedicated to the valuation aspects of land and Part 5 focuses on spatial planning information. The forthcoming Part 6 is planned to provide guidelines for the practical implementation of the LADM, emphasizing collaboration with the Open Geospatial Consortium (OGC). Ref. [5] provides a detailed overview of the latest developments of LADM Edition II. The first part of Edition II was recently published [20].

### 1.2. UN Sustainable Development Goals (SDGs) and Metadata

Global research on land indicators, exemplified by studies like [21–23], has significantly contributed to understanding the complexities of sustainable development, particularly within the framework of the SDGs. This knowledge becomes paramount when considering the extensive scope of the SDGs, as highlighted by the intricate network of 17 overarching goals, 169 targets, and 248 different indicators (of which 231 are unique), some of which are further subdivided into sub-indicators (e.g., 1.4.2 (secure land rights), 5.a.1 (women's agricultural land rights), etc.) [24,25]. The metadata for SDG indicators [26] contain vital information on the precise measurement, monitoring and reporting of progress. This includes detailed definitions of each indicator, calculation methodologies, data sources and associated indicators. For instance, the metadata for SDG 1.1, "By 2030, eradicate extreme poverty for all people everywhere, currently measured as people living on less than USD 1.25 a day", provide detailed information regarding the way "extreme poverty" is defined, the methods used for collecting and analyzing data and the measures taken to ensure that the data are comparable and consistent [27].

It is noted that the metadata of SDG indicators lack uniform standardization. The compilation of relevant documentation was carried out by different entities, resulting in deviations in the level of detail and frequency of updates. For example, Indicator 12.3.1, covering the food loss index and food waste index, has two metadata documents managed by the FAO [28] and UNEP [29]. In some cases, instead of detailed descriptions, only hyperlinks to other sources are included.

## 2. Materials and Methods

Three fundamental hypotheses underpin this ongoing research:

1. By amalgamating the standardized principles and methodologies of the ISO 19152 LADM with the overarching goals and targets of the UN Agenda 2030, the resulting land administration indicators will manifest as more comprehensive, accurate and representative.
2. The utilization of these indicators has the potential to bolster evidence-based policymaking, thereby substantively contributing to the realization of the SDGs.
3. The effectiveness and expediency of indicator computation can be significantly enhanced through continuous updates to the land administration system (LAS).

This research is divided into two main parts: first, we identify the SDGs that are related to the LADM and classify them; second, we analyze how the LADM, in the context of its second edition, applies to the SDGs. Through this paper, it is expected that the potential role and importance of the LADM in promoting sustainable land administration and supporting the achievement of the SDGs will be clearly revealed.

To facilitate this, a systematic methodology called the Four-Step Method is developed for this research, which streamlines the process from keyword extraction to the creation of a comprehensive Unified Modeling Language (UML) diagram. As shown in Figure 3, it begins with a thorough extraction and refinement of keywords, aligns these with the LADM components, categorizes the associations and culminates in the development of a detailed UML representation. This methodological framework ensures a rigorous and structured approach to analyzing and integrating land administration data.

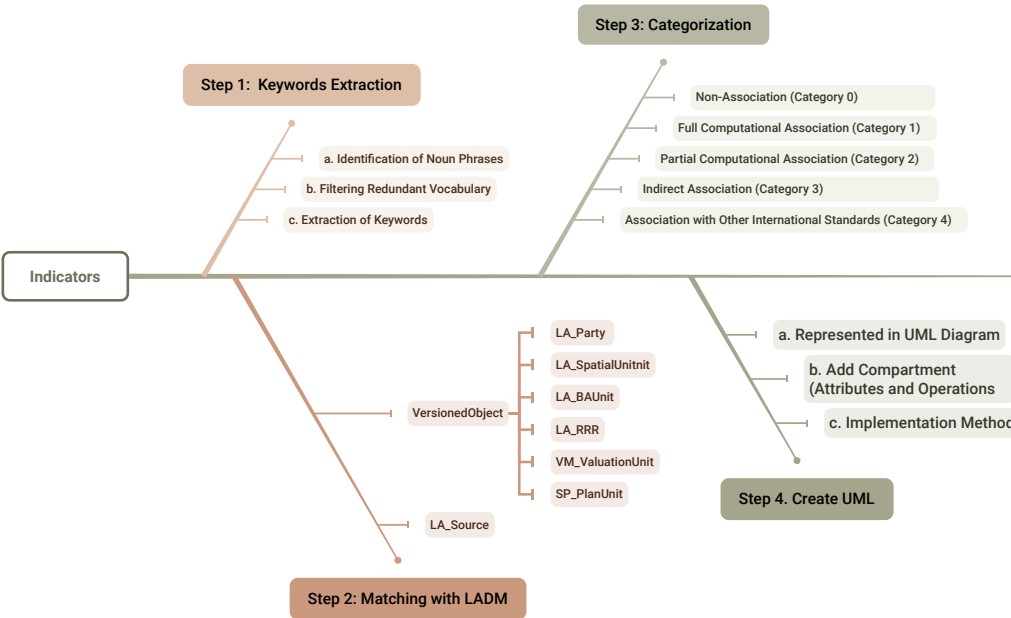

**Figure 3.** Methodology followed in this paper.

*Indicator Selection and Classification*

This section comprises four sub-sections, each one devoted to the analysis of the four steps of the methodology as presented in Figure 3.

**Step 1: Keyword Extraction and Preliminary Filtering**

First and foremost, it is important to use the core terminology of the LADM to perform an initial filtering for the identification of the relevant indicators. The core terms are Land, Party, RRRs (Rights, Responsibilities, Restrictions), Spatial Units, Marine, Valuation and Spatial Plan. The definitions of these terms are provided in the official documentation submitted to ISO/TC211 during LADM Edition II development and are briefly presented below:

1. **Land**: the spatial extent that is defined by RRRs and encompasses the surface of the earth, strata, sub-strata or the marine environment, like a building.
2. **Party**: a person or organization that plays a role in a rights, responsibilities, or restrictions transaction, like a natural person.
3. **Right**: formal or informal entitlement to own or do something.
4. **Responsibility**: formal or informal obligation on the land owner to allow or do something.
5. **Restriction**: formal or informal obligation on the land owner to refrain from doing something.
6. **Spatial Units**: the areas of land (or water, e.g., water rights and the marine environment) where the rights and social tenure relationships apply.
7. **Marine**: relating to navigation or shipping; relating to or connected with the sea; used or adopted for use at sea.
8. **Valuation**: the process of estimating the value of an immovable property.

    (a) **value**: the value of a property or a property unit estimated under certain assumptions at a particular moment in time.

9. **Spatial Plan**: a set of documents that indicates a strategic direction for the development of a given geographic area, states the policies, priorities, programs, and land allocations that will implement the strategic direction and influences the distribution of people and activities in spaces of various scales.

    (a) **plan unit**: homogenous smallest area/space (2D/3D) with an assigned function/purpose to represent the potential land use development according to the

spatial planning authorities at the highest detail and largest scale (usually the municipality/ neighborhood level)

10. **Source**: document providing legal and/or administrative facts on which the LA object [right, restriction, responsibility, basic administrative unit, party, or spatial unit] is based on.

Next, relevant words are filtered from the SDGs. The intricate semantic relationships between words and terms may involve synonymy, hypernymy or contextual relevance. Each word is subjected to careful evaluation, gauging its alignment with the core LADM terminology, thereby facilitating the creation of a semantic bridge between the landscape of the SDGs and the land administration domain. This reveals the consistency and nuances of expression between the terminology of the SDGs and the LADM, thus laying the foundation for harmonization between the two fields.

**Step 2: Matching SDGs with LADM core classes**

The selected indicators underwent a rigorous evaluation process, which involved a comprehensive analysis of their corresponding SDG indicator metadata documents [30] and a rigorous matching process. A key aspect of this evaluation was the careful examination of specific sections within the indicator metadata documents, namely, "0.f. Related indicators", "2.a. Definition and concepts", "3.a. Data sources" and "4.c. Method of Computation". Briefly:

1. **For "0.f. Related indicators"**. It identifies related indicators to understand their connections and potential overlaps, aiding in defining the evaluation scope. For example, for Indicator 5.1.1, "Whether or not legal frameworks are in place to promote, enforce and monitor equality and non-discrimination on the basis of sex", to avoid duplication, it does not cover areas of law that are addressed under Indicator 5.a.2, "Proportion of countries where the legal framework (including customary law" guarantees women's equal rights to land ownership and/or control);

2. **For "2.a. Definition and concepts"**. It provides explanations for the more generalized text used in the indicators. For example, for Indicator 1.4.1, "Proportion of population living in households with access to basic services", the precise definition of "basic services" is elaborated upon;

3. **For "3.a. Data sources"**. It gives information on the potential databases and the organizations responsible for data collection, allowing for a quick assessment of whether they are relevant to the data involved in the LADM;

4. **For "4.c. Method of Computation"**. The specific calculation methods for each indicator are detailed, which encompass a variety of approaches, like the formulation of mathematical equations, tabulated scoring systems, etc. This section also plays a pivotal role in the subsequent classification of indicators.

This thorough analysis was instrumental in ensuring the accuracy and reliability of the chosen indicators, as it provides a deep understanding of the conceptual framework and technical aspects underpinning them. Subsequently, a rigorous alignment process was conducted, associating the basic classes (and sub-classes) from the various parts of LADM Edition II with these selected indicators. As previously mentioned, the basic classes are LA_Party, LA_SpatialUnit, LA_BAUnit, LA_RRR, VM_ValuationUnit and SP_PlanUnit, and they all stem from VersionedObject (and are associated with LA_Source). For those indicators that could be matched and have explicit calculation formulas, it should be noted which data cannot be provided by the LADM and may need to be sourced from external data sets.

**Step 3: Indicator Categorization**

In the third step, a comprehensive classification of the selected indicators is executed based on their nuanced relationships with the LADM. The categorization criteria employed are:

1. **Non-Association (Category 0)**: These indicators demonstrate no discernible direct or computational correlation with the LADM.
2. **Full Computational Association (Category 1)**: Indicators falling within this category exhibit an unequivocal and comprehensive computational interdependence with the LADM. All data required for the calculation of these indicators can be obtained from a land administration system that conforms to the LADM.
3. **Partial Computational Association (Category 2)**: These indicators, while partly reliant on data provided by the LADM for their calculations, necessitate additional external data sources. They thus have a partial computational connection to the LADM.
4. **Indirect Association (Category 3)**: The LADM offers supportive roles during the indicator generation process. These roles are:

    (a) Indicator involves LADM elements (classes or attributes) but lacks direct expression (and therefore calculation) within the structure of the model. For example, for the indicator "14.6.1 Degree of implementation of international instruments aiming to combat illegal, unreported and unregulated fishing", the computation method is based on surveys and scoring, and it is related to elements such as marine and land rights within the LADM.

    (b) Indicator indirectly utilizes LADM elements, and their final expressions do not have a direct relation with the LADM. For instance, in the indicator "1.2.2 Proportion of men, women and children of all ages living in poverty in all its dimensions according to national definitions", "Poverty" includes a "housing" dimension, which is related to "BAUnit".

5. **Association with Other Standards (Category 4)**: Indicators categorized as such are fundamentally linked with other (international) standards to be computed and potentially may partly rely on the LADM.

**Step 4: Create UML**

This last step focuses on developing the indicator within the context of the LADM. The development of the indicators includes translating their computational requirements and dependencies into practical and systematic implementations; this is organized as follows:

1. **Represented in a UML Diagram**: UML class diagrams are developed to express all the information needed to calculate the indicators from the SDG metadata document, specifically:

    (a) Information that can be directly represented by packages within the current version of the LADM [5].

    (b) Information that has a well-defined source and can be linked from UML external classes to other databases. While the construction of external databases encompassing party data, address data, taxation data, land cover data, physical utility network data, earth surface data and archive data falls outside the scope of the LADM, the LADM provides stereotype classes for these data sets, which indicate what data set elements the LADM expects from these external sources.

    (c) Information that will be output in an interface class.

2. **Add compartment**: For the most relevant class, a dedicated compartment is used for the computation of the indicator values. This compartment contains the name and parameters of the added operations. It is noted that the spatial extent (country, province, municipality) and temporal extent (decade, year, month) may vary. A typical operation could be to compute an indicator value in a specific year and for a specific area, e.g., compute indicator X (year, area).

3. **Implementation Method**: For each operation, a well-defined implementation method is specified within the UML diagram. This includes an attached note defining the steps of computation. The implementation methods are articulated using programming languages (i.e., Python, Java, pseudo-code). Crucially, these methods were aligned with the information elements delineated in the utilized UML classes.

4.   **Interface Class**: For each indicator, the resulting values, as produced by the three mentioned components above, are conveniently represented in so-called interface classes. In the LADM, interface classes are used in other situations where information is collected from other classes and somehow combined, e.g., LA_SpatialUnitOverview (i.e., cadastral map) or LA_PartyPortfolio, also collecting and aggregating information from other LADM classes.

5.   **Add color**: To enhance readability, distinct colors are utilized to represent various elements: green signifies classes from the Party Package, yellow denotes classes from the Administrative Package, blue represents classes from the Spatial Unit Package, white indicates the Source Class, orange is used for classes from the Valuation Information Package, greyish blue is used for classes from the Spatial Information Package, purple signifies External Classes, brown denotes interface classes and light pink highlights methods that implement operations. Text highlighted in red emphasizes components that are crucial for the computation of the indicator.

This systematic approach ensures a structured and comprehensive documentation for indicator development and calculation in the context of the LADM, facilitating clarity and transparency in the computational processes associated with each indicator.

## 3. Results

This section delves into the practical applications and implications of linking the LADM with selected SDG indicators, exemplifying the theoretical framework established in previous sections. Through a series of case studies, the aim is to illustrate how different parts of the LADM can be effectively utilized to support and measure progress towards specific SDGs. Ranging from land registration to property valuation, each case study focuses on a different aspect of the LADM and links to a corresponding SDG indicator, thereby demonstrating the practicality and significance of the LADM in various contexts of sustainable development. The Four-Step Method is consistently employed across all cases to ensure a uniform approach to analysis.

### 3.1. LADM Part 2: Land Registration with SDG 1.4.2

The first case study explores the intersection of LADM Part 2 [5], which focuses on land registration, with SDG 1.4.2. SDG 1 aims to eradicate poverty in all its forms worldwide, with SDG 1.4 specifically targeting the promotion of secure land tenure rights for all individuals and SDG 1.4.2 serving as an indicator, measuring the proportion of the adult population with secure tenure rights to land. The Four-Step Method is applied as follows.

#### 3.1.1. Indicator Classification

*SDG Indicator 1.4.2: "Proportion of total adult population with secure tenure rights to land, (a) with legally recognized documentation, and (b) who perceive their rights to land as secure, by sex and type of tenure."*

**Step 1: Keyword Extraction**

Firstly, to make an initial judgement on whether the SDG 1.4.2 indicator is related to the LADM, the indicator description needs to be analyzed. The process of "Identification of Noun Phrases–Filtering Redundant Vocabulary–Extraction of Keywords" was used, as shown in Figure 4.

The keywords include "legally recognized documentation", "adult population", "sex", "secure tenure rights", "rights to land" and "type of tenure", and their corresponding LADM core terms are "Source", "Party" and "Rights".

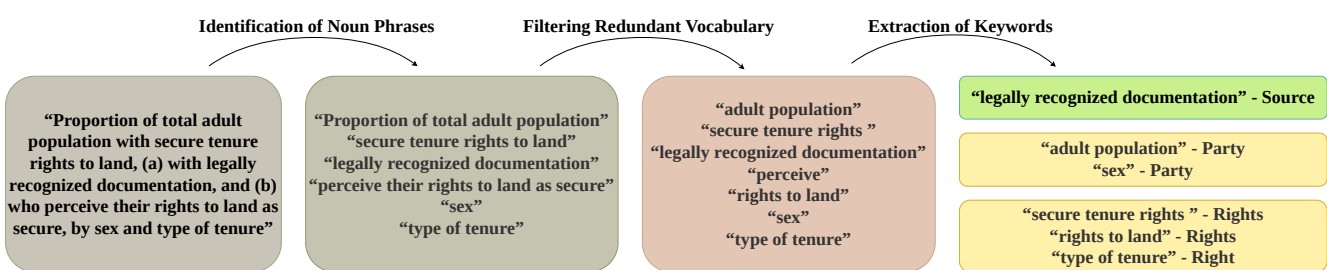

**Figure 4.** Keyword extraction for SDG 1.4.2.

**Step 2: Matching with LADM concepts**

According to the SDG Indicator 1.4.2 metadata document [8], Section "2.a. Definition and concepts", the indicator is divided into two parts:

- Part (A) quantifies the proportion of adults possessing legally recognized documentation over land within the total adult population.
- Part (B) emphasizes the proportion of adults who perceive their land rights as secure within the adult population.

Mathematically, these parts are represented as follows:

$$\text{Part(A)}: \frac{\textit{People (adult) with legally recognized documentation over land}}{\textit{Total adult population}} \times 100$$

$$\text{Part(B)}: \frac{\textit{People (adult) who perceive their land rights as secure}}{\textit{Total adult population}} \times 100$$

Therefore, to meet the metadata documentation requirements, it is necessary to collect three types of data:

1. The number of adults possessing legally recognized documentation over land (for Part A), which can be derived from a land administration system compliant with the LADM;
2. The number of adults who perceive their land rights as secure (for Part B), which can only be obtained through household surveys (or relevant historical data stored in external databases);
3. The total adult population (for both), made available from censuses or inter-censual projections.

Then, based on the analysis carried out, the following conclusions were drawn:

1. "Secure tenure rights" and "type of tenure" are related to the LA_RRR (Rights, Responsibilities, Restrictions) class due to their connection to the nature of land rights.
2. "Legally recognized documentation", as mentioned in the metadata file, is legal documentation of rights that refers to the recording and publication of information on the nature and location of land, rights and right holders recognized by the government. So, it is related to LA_Source.
3. "Sex" should be an attribute of LA_Party [31].

**Step 3: Categorization**

SDG Indicator 1.4.2 (a) is classified under the "Partial Computational Association (Category 2)". SDG Indicator 1.4.2 (b) is categorized under the "Indirect Association (Category 3)". So, SDG Indicator 1.4.2 is classified under the "Partial Computational Association (Category 2)".

3.1.2. Indicator Development (Step 4: Create UML)

In accordance with the indicator development steps outlined in Section 2, the computation of SDG Indicator 1.4.2 is depicted through a UML diagram. The specific steps in the modeling process for SDG Indicator 1.4.2 are as follows:

1. **Represented in the UML Diagram:** The core classes of the LADM (LA_Party, LA_RRR, LA_BAUnit, and LA_Source) in the UML diagram describing the indicator computation are introduced. To fulfill the requirements of SDG Indicator 1.4.2, an external class named "ExtSecureLandRightAdult" is introduced to represent the molecular aspect of Part B and an external class named "ExtParty".

2. **Add Compartment (Attributes and Operations):** Compartments for the aforementioned classes are created and interconnected within the UML diagram, with each compartment incorporating specific attributes and operations. Detailed descriptions are shown in Tables 2 and 3.

3. **Implementation Method:** Having completed the addition of attributes and operations, the third step is concerned with developing a method for the logic required to calculate the indicator to ensure that the previously established theoretical framework is translated into workable algorithms. This includes the methods "computeProportionWithLegalDocumentation", which calculates the proportion of adults with legally recognized land rights documentation; "computeProportionPerceivingSecurity", which computes the proportion of adults perceiving land rights as secure; "generateReport", which synthesizes data into a comprehensive report; and "countAdults", which determines the total count of adults. The final UML diagram for SDG 1.4.2 is shown in Figure 5.

**Table 2.** Attributes added to SDG 1.4.2 UML from an existing LADM class.

| Existing LADM Class | Attributes Used in the Case | Notes |
| --- | --- | --- |
| LA_Party | "+gender:LA_HumanSexesType[0..1]" | Highlighted to facilitate gender-based classification and calculation [31,32]. |
| LA_Right | "LA_RightType" | Delineates various land tenure types, echoing the "type of tenure" parameter in the indicator. The specific right types are detailed in the "Code List". |
| LA_AdministrativeSource | "+ type: LA_AdministrativeSourceType" | Signifies "Legally recognized documentation". Its codelist meticulously enumerates the possible value, like agriLease, deed and title. |
| LA_BAUnit | | While not the focal point, it is outlined to underscore the indicator's emphasis on rights over land. |

**Table 3.** New classes added for SDG 1.4.2 include attributes and operations.

| New Class | Attributes Used in the Case | Operations | Notes for Class |
| --- | --- | --- | --- |
| External::ExtParty, | "+ birthday: Date" | "countAdult", facilitates the determination of the total adult count | Associated with national population databases; emphasizes the attribute birthday for determining adulthood. |
| External::ExtPartyPerceiveSecureLandRights, | "+ selfPerception: ExtLandRightPerception[0..1]" | | Primarily sourced from household surveys.<br><br>"1" indicates perceived security of land rights, and "0" indicates insecurity. |

| New Class | Attributes Used in the Case | Operations | Notes for Class |
|---|---|---|---|
| | "+ totalAdult Population: Integer" | | The total adult population of a given region in a given year. |
| | | | These data are obtained from the External::ExtParty class, specifically from the birthday attribute, using the "countAdult" operation to calculate the total number of adults in an area. |
| | | | Has a pivotal role by encapsulating the final computations of the indicator values. It operates as an aggregated construct, amalgamating and interpreting data sourced from the various classes and external entities. |
| Interface:: SDG_1.4.2 | "+ adultsWithLegal Documentation: Dictionary <GenderTenureKey, Integer>" | | Aggregated information from LA_Party, LA_Right, and LA_AdministrativeSource classes. |
| | | | The LA_Party class provides information about parties (individuals or organizations), the LA_Right class represents legal rights associated with land and the LA_AdministrativeSource class provides information about the legal recognition of such rights through documents. Together, these classes help determine which adults have legally recognized documentation of land rights, categorized by gender and type of tenure. |
| | | | "GenderTenureKey" is a composite key used in the "Dictionary<GenderTenureKey, Integer>". It consists of "+ gender: LA_HumanSexesType" from LA_Party and "+ rightType: LA_RightType" from LA_Right. |
| | "+ adultsPerceiving SecureRights: Dictionary <GenderTenureKey, Integer>" | | Aggregated information from External::ExtPartyPerceiveSecureLandRights and External::ExtParty classes. |
| | | | The former provides information on those who consider their land rights to be secure (i.e., + selfPerception = 1) and the latter provides information on adults. Combined, they help to summarize the adults who consider their land rights to be secure, disaggregated by gender and tenure type. |
| | | "+computeProportion WithLegalDocument ation(year, area, gender, tenureType): Float" | This operation calculates the proportion of adults possessing legally recognized documentation for land rights, factoring in variables such as spatial extent, temporal context, gender and type of land tenure. |

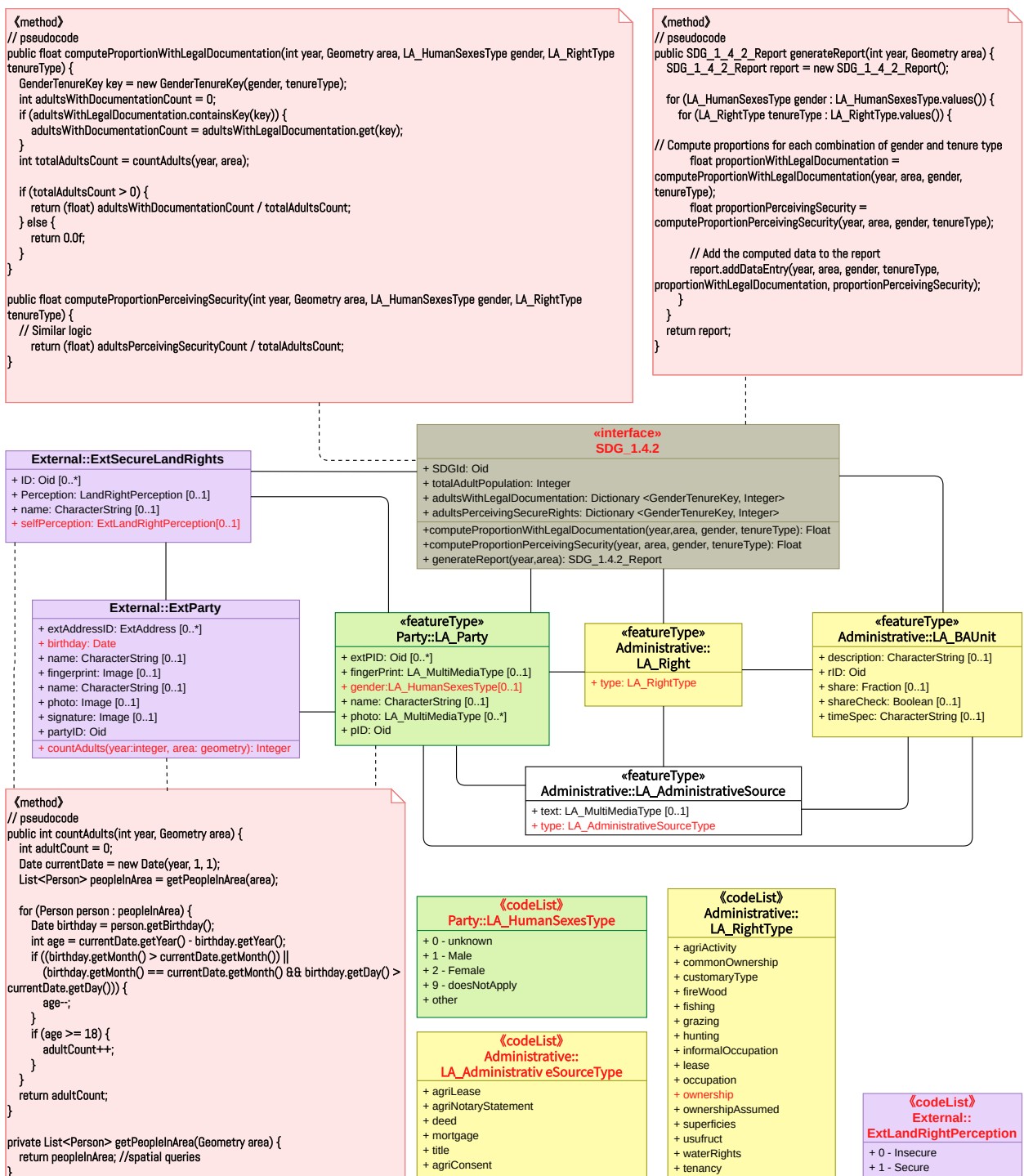

**Figure 5.** Modeling of SDG Indicator 1.4.2 calculation in a UML class diagram.

### 3.2. LADM Part 2: Land Registration and SDG 5.a.1

Before delving into the specifics of SDG Indicator 5.a.1, it is worth noting the close linkage between SDGs 1.4.2 and 5.a.1. Both indicators share a common focus on land tenure and security, albeit within different gender perspectives. SDG 5 aims to achieve gender equality and empower all women and girls. Specifically, SDG 5.a.1 focuses on ensuring women's equal rights to land ownership and tenure security. Consequently, the UML diagram developed for SDG 1.4.2 provides a foundational framework that can be extended to SDG 5.a.1, demonstrating the versatility and efficiency of the LADM in addressing diverse SDGs while promoting gender equality and women's empowerment.

3.2.1. Indicator Classification

*SDG Indicator 5.a.1: "(a) Proportion of total agricultural population with ownership or secure rights over agricultural land, by sex; and (b) share of women among owners or rights-bearers of agricultural land, by type of tenure."*

**Step 1: Keyword Extraction**

Similar to the previous case, the process is initiated by identifying and extracting key terms from the SDG indicator. The keywords include "agricultural population", "ownership", "secure rights", "agricultural land," "sex" and "type of tenure". Corresponding LADM terms include "LA_Party", "LA_RRR" and "LA_BAUnit".

**Step 2: Matching with LADM concepts**

In the SDG Indicator 5.a.1 metadata document, according to the "2.a. Definition and concepts" section, this indicator also consists of two sub-indicators:

- Sub-indicator 5.a.1 (a): measurement of the proportion of the agricultural population with ownership or secure rights over agricultural land, disaggregated by sex, is intended to reveal the distribution of men and women in terms of their rights and interests in agricultural land.
- Sub-indicator 5.a.1 (b) focuses on gender parity and reflects the current status of gender disparities in agriculture by measuring whether women are disadvantaged relative to men in terms of access to ownership or secure rights over agricultural land.

Mathematically, these two sub-indicators are represented as follows:

$$(a): \frac{\textit{No. of people in agricultural population with ownership or secure rights over agricultural land}}{\textit{Total agricultural population}} \times 100 \textit{ by sex}$$

$$(b): \frac{\textit{No. of women in agricultural population with ownership or secure rights over agricultural land}}{\textit{Total in the agricultural population with ownership or secure rights over agricultural land}} \times 100 \textit{ by type of tenure}$$

There are two new data needed to compute SDG Indicator 5.a.1:

1. The number of people in the agricultural population. This refers to the population of adult individuals living in agricultural households. While the LADM has the class LA_GroupParty to represent group party, it lacks a specific attribute for "agricultural family". Consequently, these data need to be sourced externally, such as from agricultural surveys, general household surveys (GHSs) or agricultural censuses. This information can be effectively represented using the External::ExtParty Class or by expanding the LA_GroupPartyType codelist to include a value for "agricultural family".

2. The number of people with ownership or secure rights over agricultural land. For "agricultural land", the metadata specify that "Land is considered 'agricultural land' according to its use". LA_ParcelUseType is a code list in the LADM spatial unit package that corresponds to "agricultural land" and contains an "agricultural" value to describe parcels of land that are used for agricultural purposes. For "ownership or secure rights", considering the diversity in land ownership systems across countries and the need for comparability, the metadata specify that "to determine whether an individual is said to have ownership or secure rights to agricultural land three conditions (proxies) are considered: Formal documentation: Proxy 1—Presence of legally recognized documents in the name of the individual; Alienation rights: Proxy 2—Right to sell and Proxy 3—Right to bequeath". Specifically,

   (a) Formal documentation: This is the existence of any document that an individual can use to claim property rights before the law over an asset by virtue of the individual's name being listed as owner/co-owner or holder/co-holder on the document. Correspondingly, within the administrative package of the LADM,

there is a code list named LA_AdministrativeSourceType, which represents the type of document, such as title, deed, agricultural lease, etc.

(b) Alienation rights: In the absence of formal written documentation, rights to sell and bequeath are considered objective facts that carry legal force as opposed to a simple self-reported declaration of tenure rights over land. This implies the necessity for a legal context to support these rights. Although LADM Part 2 does not explicitly have a specialized class for legal content, LADM Part 3 includes a new object: the "Governance". The class MG_Governance has been included to allow for the description of the context information from a proclamation, law or treaty document. This addition enriches the administrative structure by allowing legal texts to be associated with an administrative unit. Consequently, to address this need within the LADM, it would be feasible to introduce a class akin to "LA_Governance". This class would serve to represent the legal context and governance information, providing a structured representation of legal frameworks relevant to land rights.

**Step 3: Categorization**

SDG Indicator 5.a.1 is classified under the"Partial Computational Association (Category 2)". For the indicator, all data are available from the LADM, except for the agricultural population, which requires an additional data source.

### 3.2.2. Indicator Development (Step 4: Create UML)

1.  **Represented in a UML Diagram:** The UML diagram for 5.a.1 builds upon the structure established for 1.4.2, with enhancements to accommodate the specifics of agricultural land tenure and legal context. Notable additions include LA_SpatialUnit and its subclass LA_LegalSpaceParcel for identifying agricultural land, LA_Governance for legal context and legal information, and ExtAgriculturalHouseholds for details related to agricultural households. To distinguish these new components, the classes used only in 1.4.2 are grey, and those used in 5.a.1 are colored.
2.  **Add Compartment (Attributes and Operations):** According to Table 4, the modifications or additions to the 1.4.2 UML diagram are required; otherwise, they remain unchanged.

**Table 4.** Classes with attributes and operations for SDG 5.a.1.

| Class | Attributes Used in the Case | Operations | Notes for Class |
|---|---|---|---|
| LA_LegalSpaceParcel | "+ type: LA_ParcelUseType [0..*]" | | LA_ParcelUseType is a codelist, which includes "agricultural" to denote parcels used for agriculture. |
| LA_Governance | "governanceTitle" | | Provide the title of the governance document |
| | "governanceDescription" | | Detail the governance statement |
| External::ExtAgricultural-Households | "+ InAgriculturalHouseholds: Boolean[0..1]" | | To ascertain if an individual is part of an agricultural household. |
| External::ExtParty | | "+ countAgriculturalPopulation(year, area): integer" | To determine the total agricultural population within a specified period and area. |

**Table 4.** *Cont.*

| Class | Attributes Used in the Case | Operations | Notes for Class |
|---|---|---|---|
| Interface Class:: SDG_1.4.2&5.a.1 | "+ totalAgriculturalPop: Integer" | | Derived from ExtParty |
| | "+ peopleWithOwnershipOrSecureRightsOnAgriculturalLand: Dictionary<GenderTenureKey, Integer>" | | To aggregate information from the classes LA_Party, LA_Right, LA_BAUnit, LA_SpatialUnit and LA_Governance |
| | | "+ computeProportionWithOwnershipOrSecureRightsOnAgriculturalLand (year, area, gender): Float" | To calculate the proportion of the agricultural population with ownership or secure rights over agricultural land, disaggregated by gender |
| | | "+ computeWomenProportionWithOwnershipOrSecureRightsOnAgriculturalLand(year, area, tenureType): Float" | To assess the share of women among those with ownership or secure rights over agricultural land, disaggregated by tenure type. |
| | | "+ generateReport(year, area): SDG_1.4.2&5.a.1_Report" | To synthesize the computed data into a comprehensive report. |

3. **Implementation Method:** The methods include "+ countAgriculturalPopulation(year, area): integer" from the ExtParty class and various operations within the SDG_1.4.2 and 5.a.1 interface class. Table 5 has a detailed explanation of every method. The final UML is shown in Figure 6.

**Table 5.** Method explanation for SDG 5.a.1.

| Method | Purpose | Implementation Steps | Details |
|---|---|---|---|
| + countAgriculturalPopulation(year, area): integer | Calculates the total number of individuals in agricultural households within a specified geographical area and time frame. | i. Data Retrieval | Collect data from agricultural surveys, census data or agricultural household databases. |
| | | ii. Data Filtering | Filter the data to include only individuals classified as part of the agricultural population. |
| | | iii. Counting | Count the number of individuals meeting the criteria within the specified year and area. |

| Method | Purpose | Implementation Steps | Details |
|---|---|---|---|
| + computeProportionWithOwnershipOrSecureRightsOnAgriculturalLand (year, area, gender): Float | Computes the proportion of the agricultural population with ownership or secure rights over agricultural land, segmented by gender. | i. Retrieve data | Gather information on land ownership and secure rights from LA_Party, LA_Right, LA_BAUnit, LA_SpatialUnit, LA_Governance, and ExtParty. |
| | | ii. Match Criteria | Identify individuals who meet the criteria for having ownership or secure rights over agricultural land. |
| | | iii. Segment by Gender | Use the gender attribute from LA_Party to segment the data by gender. |
| | | iv. Calculate Proportion | Divide the count of individuals with ownership or secure rights by the total agricultural population for the specified area and year, obtained from the method above. |
| + computeWomenProportionWithOwnershipOrSecureRightsOnAgriculturalLand(year, area, tenureType): Float | Calculates the proportion of women within the agricultural population who have ownership or secure rights over agricultural land, categorized by tenure type. | i. Retrieve and Filter Data | Similar to the previous operation but focus on data related to women. |
| | | ii. Match Tenure Type | Use the tenure type information from class LA_Right. |
| | | iii. Calculate Proportion | Compute the proportion of women with ownership or secure rights over agricultural land out of the total number of individuals with such rights. |
| + generateReport(year, area): SDG_1.4.2&5.a.1_Report | Synthesizes the computed data into a comprehensive report, providing a holistic view of land tenure security and gender disparities in the agricultural sector. | i. Invoke Computations | Use the four methods mentioned above to get the required metrics. |
| | | ii. Format Report | Organize the output data from the four functions of 1.4.2 and 5.a.1 into a structured report format. |

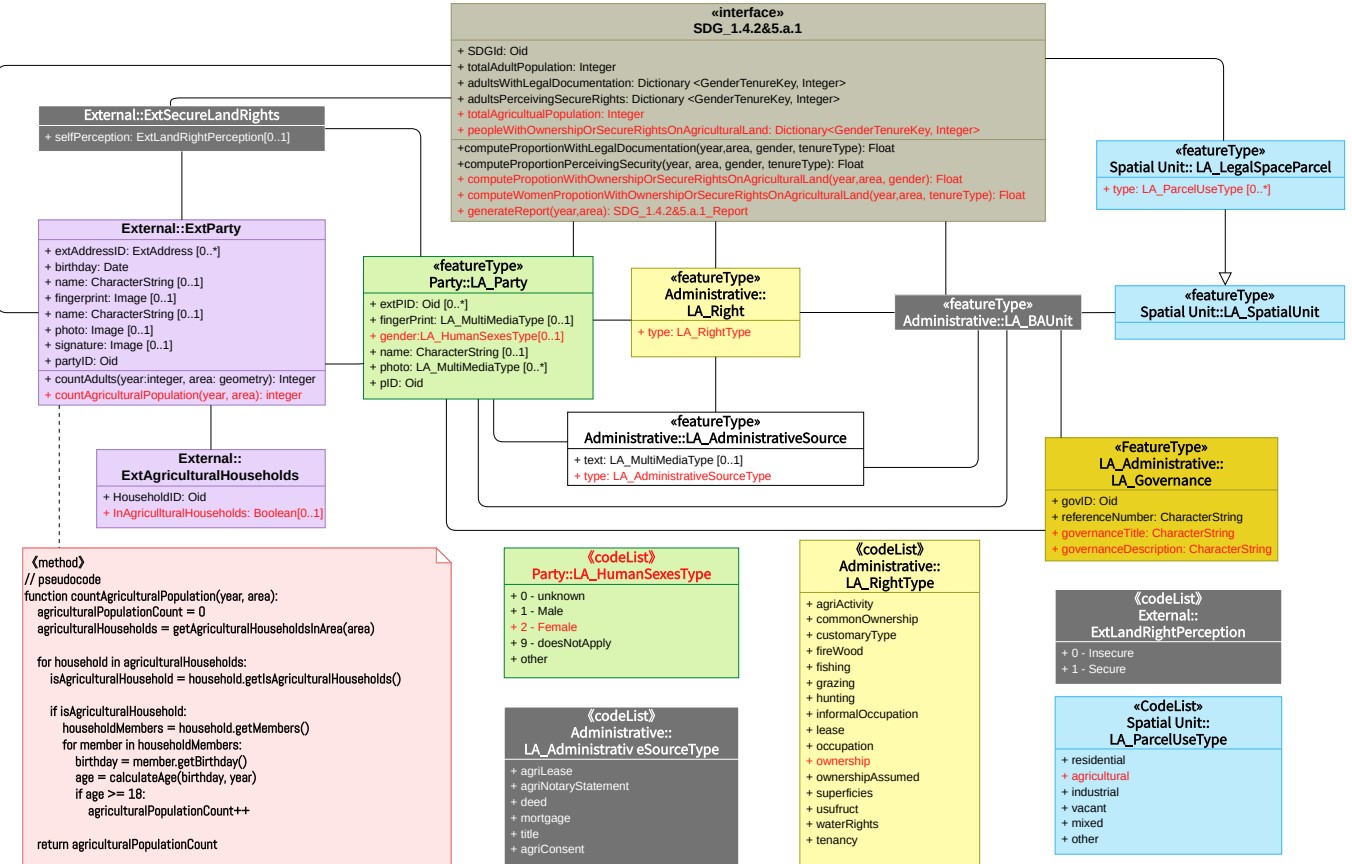

**Figure 6.** Modeling of SDG Indicator 1.4.2 and 5.a.1 calculation in a UML class diagram.

### 3.3. LADM Part 3: Marine Georegulation and SDG 14.5.1

LADM part 3 addresses georegulation in the marine environment, acknowledging the varying rights and obligations that may exist at different levels: on the surface, in the water column, and on the seabed. The model defined in Part 3 may be used for marine cadastres as well as other use cases such as Conservation Areas, Living Resources and Fishery Management Areas, Non-Living Resources Management Areas, Seabed tenure, etc.

The third case study focuses on protected areas, aiming to assess the extent and effectiveness of protected areas. This study aligns with SDG 14, which aims to conserve and sustainably use the oceans, seas, and marine resources. The case study involves aligning relevant classes and attributes within the LADM framework with SDG 14 indicators, particularly focusing on how marine areas and protected zones are represented and managed within the system.

#### 3.3.1. Indicator Classification

*SDG Indicator 14.5.1 "Coverage of protected areas in relation to marine areas."*

**Step 1: Keyword Extraction**

Given the concise nature of the definition for SDG Indicator 14.5.1, the extraction of key terms is straightforward. The term "protected area" is associated with "Spatial Units", and "marine area" correlates with "Marine".

**Step 2: Matching with LADM concepts**

In the metadata's "2.a. Definition and Concepts" section, protected areas are delineated as per the IUCN definition ([33]) as "clearly defined geographical spaces, recognized, dedicated, and managed, through legal or other effective means, to achieve the long-term conservation of nature with associated ecosystem services and cultural values." Within the context of the LADM, these "clearly defined geographical spaces" correspond to the class

MG_SpatialUnit::MG_Zone, which can represents two-dimensional polygons delineating the spatial extent of protected areas. Furthermore, the notion of "legal or other effective means", referring to "national legislation or common practice (e.g., by means of an executive decree or the like)" is represented by the MG_Governance class in the LADM. This class is an added object to the administrative structure to house legal texts associated with an administrative unit, thereby providing a structured and legally recognized representation.

The computation of this indicator, as detailed in the metadata's "4.c. Method of Computation", involves a complex methodology, formulated as follows: The intersection of Zone(KBA) with Zone(Ocean) over Zone(KBA), if greater than 5%, defines Zone(marine KBAs). Subsequently, the intersection of Zone(Marine KBAs) with Zone(protected area) over Zone(Marine KBAs) calculates the percentage of protected area for every marine KBA. The summation of all protected Zone(Marine KBAs) over the number of Zone(Marine KBAs) yields the indicator value.

$$\frac{\text{Zone(KBA)} \cap \text{Zone(Ocean)}}{\text{Zone(KBA)}} > 5\% \Rightarrow \text{Zone(marine KBAs)}$$

$$\frac{\text{Zone(Marine KBAs)} \cap \text{Zone(protected area)}}{\text{Zone(Marine KBAs)}} => \text{for every mKBA, \% protected area}$$

$$\text{Sum(all Zone(Marine KBAs) protected)}/\text{Num(Zone(Marine KBAs))} => \text{the indicator}$$

This calculation relies on digital polygons:

1. Data on protected areas are sourced from the World Database on Protected Areas (WDPA), and the World Database on OECM (WDOECM). Although these databases provide well-organized information, theoretically, similar data can be derived from the LADM. Notably, the LADM can offer high-precision two-dimensional polygon files, retain pertinent legal information, and extend to three-dimensional volume files, paving the way for future expansions.
2. Data on Key Biodiversity Areas (KBAs), sourced from the World Database on Key Biodiversity Areas (WDKBA). It is worth noting that this specific data set cannot be directly obtained from the LADM.
3. Ocean area: In LADM Part 3, MG_Zone includes the attribute "+ zoneObjectType", with a code list MG_ZoneTypeList. The code list describes categories that have a common characteristic related to the legal and administrative aspects of the marine environment.

**Step 3: Categorization**

SDG Indicator 14.5.1 is classified under the"Partial Computational Association (Category 2)". For the indicator, all data are available from the LADM, except for data on Key Biodiversity Areas (KBAs).

3.3.2. Indicator Development (Step 4: Create UML)

In developing the SDG 14.5.1 indicator under LADM Part 3, the focus is on accurately representing and managing marine protected areas, both legally and spatially. This involves two key aspects: firstly, the legal governance of these areas, and secondly, their spatial delineation. While current models primarily reflect two-dimensional spatial extents, the LADM's flexible structure allows for future expansion into three-dimensional representations.

**1 and 2. Represented in a UML Diagram and Add Compartment**

The main components of the UML diagram constructed for SDG 14.5.1 are shown in Tables 6 and 7.

**Table 6.** Attributes added to SDG 14.5.1 UML from an existing LADM class.

| Existing LADM Class | Attributes Used in the Case | Notes |
| --- | --- | --- |
| MG_Zone | "+ zoneObjectType" | Provides the maritime limits and boundaries. |
| MG_Surface | "+ geometry" | Provides two-dimensional spatial geometry for marine areas. |
| MG_Governance | "govID", "governanceTitle", and "governanceDescription" | Provides the legal context and framework for marine protected areas. |
| MG_FeatureUnit | | Represents features such as marine areas and protected areas. |
| MG_SpatialAttributeType | "+ geometry" | Provides two-dimensional spatial geometry for protected area. |

**Table 7.** New classes added for SDG 14.5.1 include attributes and operations.

| New Class | Attributes | Operations | Notes |
| --- | --- | --- | --- |
| External::ExtKeyBiodiversityArea | "+ geometry" | | Connects with external databases to integrate data on Key Biodiversity Areas (KBAs). |
| Interface: SDG_14.5.1 | "+ MarineArea" "+ KBA" "+ protectedArea" | "+ getMarineKBAs(): List<Geometry>" "+ calculateOverallProtection-Ratio(): Float" | Serves as the center for integrating data |

## 3. Implementation Method

In this section, the methods to implement the "+ calculateOverallProtectionRatio" and "+ getMarineKBAs" operations, which are integral to the interface class's ability to quantify and retrieve key data related to Marine Protected Areas, are described. Detailed descriptions are listed below and the final UML is shown in Figure 7.

1. + getMarineKBAs(): List<Geometry>: This operation is dedicated to identifying and retrieving all zones classified as Marine Key Biodiversity Areas (marine KBAs). These zones are discerned by computing the intersection of marine areas (MarineArea) with Key Biodiversity Areas (KBAs), followed by filtering based on the criterion that the intersected area constitutes over 5% of the total area of the KBA.

    (a) Data Acquisition: Extract all KBA zones (sourced from + KBA: Geometry) along with the comprehensive marine area (+ MarineArea: Geometry).

    (b) Intersection Computation: For each KBA zone, calculate its spatial intersection with the marine area.

    (c) Percentage Calculation: Ascertain the percentage of the intersected area relative to the total area of the KBA.

    (d) Identification of marine KBAs: Filter to select those zones where the intersected area exceeds 5% of the total KBA area, designating them as marine KBAs.

    (e) Output: Generate a list of geometrical objects representing all identified marine KBAs.

2. + calculateOverallProtectionRatio(): Float: This operation computes the ratio of the total area designated as protected within the marine KBAs to the overall area of marine KBAs.

    (a) Data Acquisition: Employ the getMarineKBAs() method to amass all marine KBAs and retrieve the geometrical representation of protected areas (+protectedArea: Geometry).

(b)     Protected Area Computation: For each marine KBA, determine the intersection with the protected area and cumulatively sum the area of these intersections to represent the total protected area.

(c)     Total Area Computation: Calculate the aggregate area encompassing all marine KBAs.

(d)     Ratio Computation: Establish the ratio of the total protected area to the total area of marine KBAs.

(e)     Output: Yield the calculated ratio, signifying the overall coverage rate of protected zones within marine KBAs.

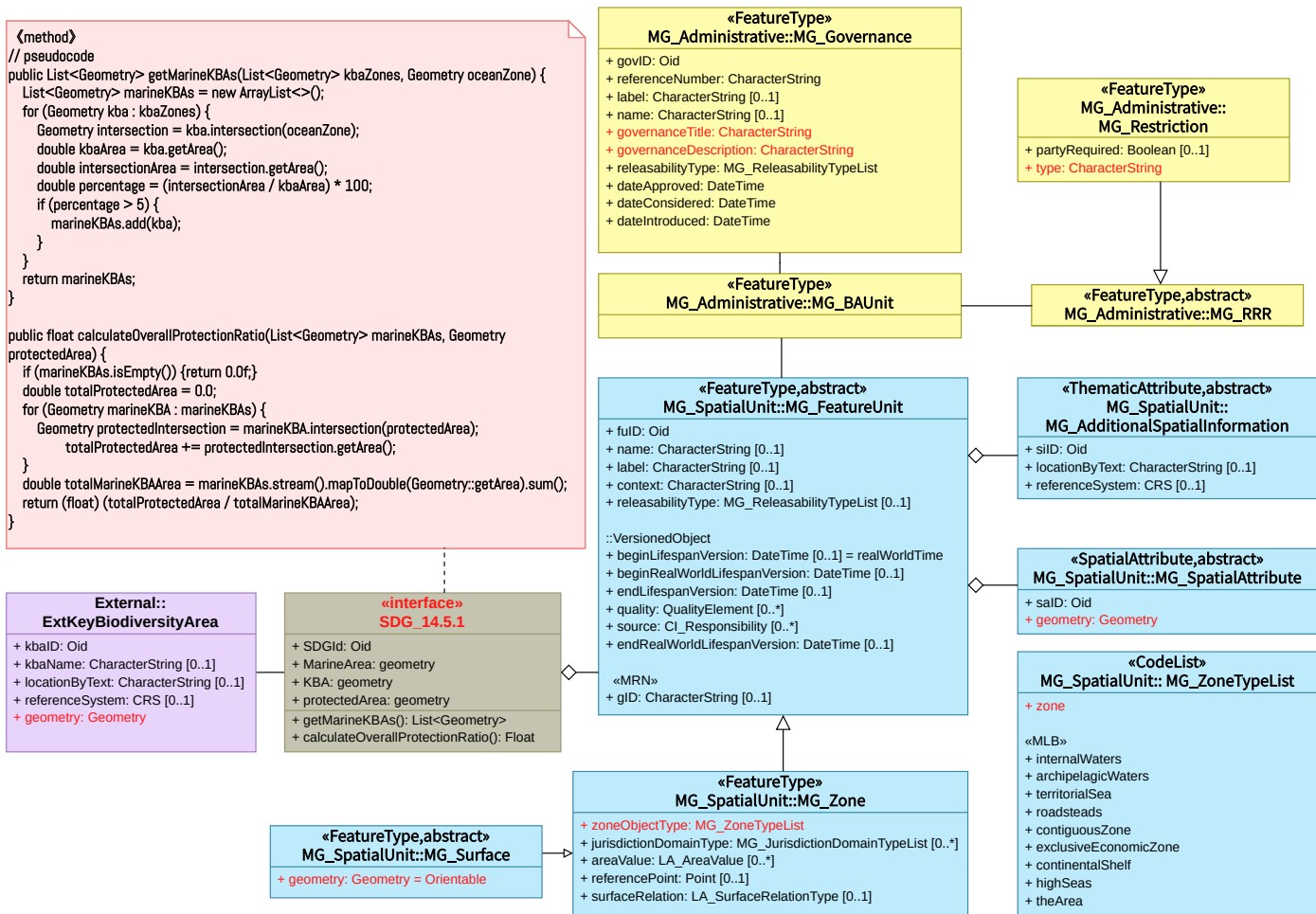

**Figure 7.** Modeling of SDG Indicator 14.5.1 calculation in a UML class diagram.

### 3.4. LADM Part 4: Valuation Information and SDG 11.5.2

Part 4 of the LADM, on the valuation of land and property, recognizes the intricate interplay between economic valuation and the physical world by providing a structured approach to capturing the value of land parcels and buildings.

The fourth case study explores the relationship between direct economic losses from disasters and global GDP through SDG indicators 11.5.2. SDG 11 aims to make cities and human settlements inclusive, safe, resilient and sustainable, which emphasizes disaster risk reduction and resilience-building efforts within urban areas. In this case study, attention is given to quantifying economic losses within the LADM framework, refining how units of valuation, transaction prices and total values can be integrated to provide a comprehensive assessment of economic losses in the context of urban resilience and sustainable development.

3.4.1. Indicator Classification

*SDG Indicator 11.5.2 "Direct economic loss attributed to disasters in relation to global gross domestic product (GDP)".*

**Step 1: Keyword Extraction**

SDG indicators 11.5.2 are the same and the extracted keyword is "Direct economic loss" corresponding to "Valuation" in the LADM.

**Step 2: Matching with LADM concepts**

For SDG Indicator 11.5.2, the metadata document gives the basic formula and concept of "direct economic loss"; more detailed information is in the technical guide. This section will explore the relevance of each component of Equation (Cx) to the LADM by referring primarily to the specific methodology in the technical guide. The related indicators as of February 2020 are:

$$X = \frac{(\mathbb{C}2 + \mathbb{C}3 + \mathbb{C}4 + \mathbb{C}5 + \mathbb{C}6)}{\text{Global GDP}}$$

where X represents the direct economic loss from disasters as a proportion of global GDP. It quantifies the monetary value of the damage to physical assets in the affected area. This includes infrastructure like homes, schools and hospitals, as well as business assets and agricultural production. Direct economic losses, typically occurring during or shortly after the event, are assessed to estimate recovery costs and claim insurance payments. This progress corresponds to the "VM_Valuation" class in LADM Part 4's Valuation Information Package, representing the output of a valuation process. The term "monetary value" parallels "assessedValue", an attribute of the "VM_Valuation" class, and the term "physical assets" parallels "spatial units", which in the valuation context, is denoted as "VM_ValuationUnit" within the Valuation Information Package.

Specifically, X is composed of components C2 to C6, representing direct agricultural loss (C2), all other damaged or destroyed productive asset losses (C3), housing sector loss (C4), critical infrastructure loss (C5) and cultural heritage loss (C6), all attributed to disasters.

In the LADM, each category of direct economic loss can be effectively mapped to specific classes and attributes.

1. C2 (Direct Agricultural Loss): This encompasses losses in both agricultural products and productive assets. In the LADM, this can be correlated with VM_ValuationUnit, which represents the basic administrative units of valuation registries. It covers various objects of valuation, including land parcels (VM_SpatialUnit) and buildings (VM_Building) that are essential in agricultural settings.
2. C3 (Economic Loss to Productive Assets): This category covers losses to assets in various economic sectors, with the LADM specifically detailing information for physical assets like industrial, office and trade buildings. This is identified using the "useType" attribute in the VM_CondominiumUnit class from the LADM.
3. C4 (Housing Sector Loss): This relates to residential buildings, which are directly classified under the "residential" category in the VM_CondominiumUnit class of the LADM.
4. C5 (Loss to Critical Infrastructure): This includes losses to essential buildings, roads and other infrastructure. The buildings part of the critical infrastructure can be identified under the "publicService" category in VM_Building's "useType". For road-related losses, the "currentLandUse" attribute of VM_SpatialUnit in the LADM can be used, with the possibility of being classified as "road".
5. C6 (Loss to Cultural Heritage): Although Part 4 of the LADM does not have a specific class for cultural heritage, the LA_Restriction class can provide insights. Protected areas, often encompassing cultural heritage sites, can be inferred from LA_Restriction's "protected" attribute.

**Step 3: Categorization**

SDG Indicator 11.5.2 is classified under the "Partial Computational Association (Category 2)". For the indicator, all data are available from the LADM, except for the global GDP, a figure that must be sourced from external data.

3.4.2. Indicator Development (Step 4: Create UML)

**1 and 2. Represented in a UML Diagram and Add Compartment**

The main components of the UML diagram constructed for SDG 11.5.2 are listed in Tables 8 and 9.

**Table 8.** Attributes added to SDG 11.5.2 UML from an existing LADM class.

| Existing LADM Class | Attributes Used in the Case | Notes |
|---|---|---|
| VM_SpatialUnit | "+currentLandUse" | Identifies land parcels used for agriculture (C2), and roads (C5) |
| VM_Building | "+ useType: VM_Building/CondominiumUseType" | Represents agricultural buildings as productive assets (C2), other productive assets (C3), residential buildings (C4) and buildings serving public services (C5). |
| LA_Restriction | "+ type: LA_RestrictionType" | Identifies the land where the cultural heritage site is located based on the value "protection" (C6). |
| VM_ValuationUnit | | Includes the basic administrative units of valuation registries, functioning as a nexus for diverse objects of valuation and a link to the valuation process classes. |
| VM_Valuation | "+ assessedValue" | Specifies the monetary valuation of assets. |
| VM_TransactionPrice | | Historical transaction prices, which can serve as a benchmark for current valuations, offering a referential perspective for comparative loss assessments. |

**Table 9.** New classes added for SDG 11.5.2 include attributes and operations.

| New Class | Attributes | Operations | Notes |
|---|---|---|---|
| External::ExtGDP | "+ gpdValue: Currency" | | Provides the specific values of global GDP in a certain year. |
| Interface: SDG_11.5.2 | "globalGDP: Currency" "+ valuationData: List<VM_Valuation>" | "+ calculateDirectEconomicLoss(category: String): Currency" "+ calculateTotalDirectEconomicLoss(): Currency" "+ calculateDirectEconomicLossRatio(): Float" "+ generateEconomicLossReport(): SDG_1.5.2_Report" | |

**3. Implementation Method**

In this case, all of the operations are centralized in the interface class, specifically:

1. "+ calculateDirectEconomicLoss(category: String): Currency": This method calculates the direct economic loss for a specified category of assets, such as agriculture, housing or infrastructure, post-disaster. It leverages the valuation information within the VM_ValuationUnit to quantify the damage in monetary terms.

2. "+ calculateTotalDirectEconomicLoss(): Currency": The summation is predicated on the outputs from the " + calculateDirectEconomicLoss" method for each category.

3.  "+ calculateDirectEconomicLossRatio(): Float": This method computes the ratio of the total direct economic loss to the global GDP. It requires interfacing with the ExtGDP class to retrieve the global GDP value and utilizes the total direct economic loss computed by the calculateTotalDirectEconomicLoss method.

4.  "+ generateEconomicLossReport(): SDG_1.5.2_Report": A comprehensive synthesis method that amalgamates all computed data and contextual information into a report. And the final UML diagram is shown in Figure 8.

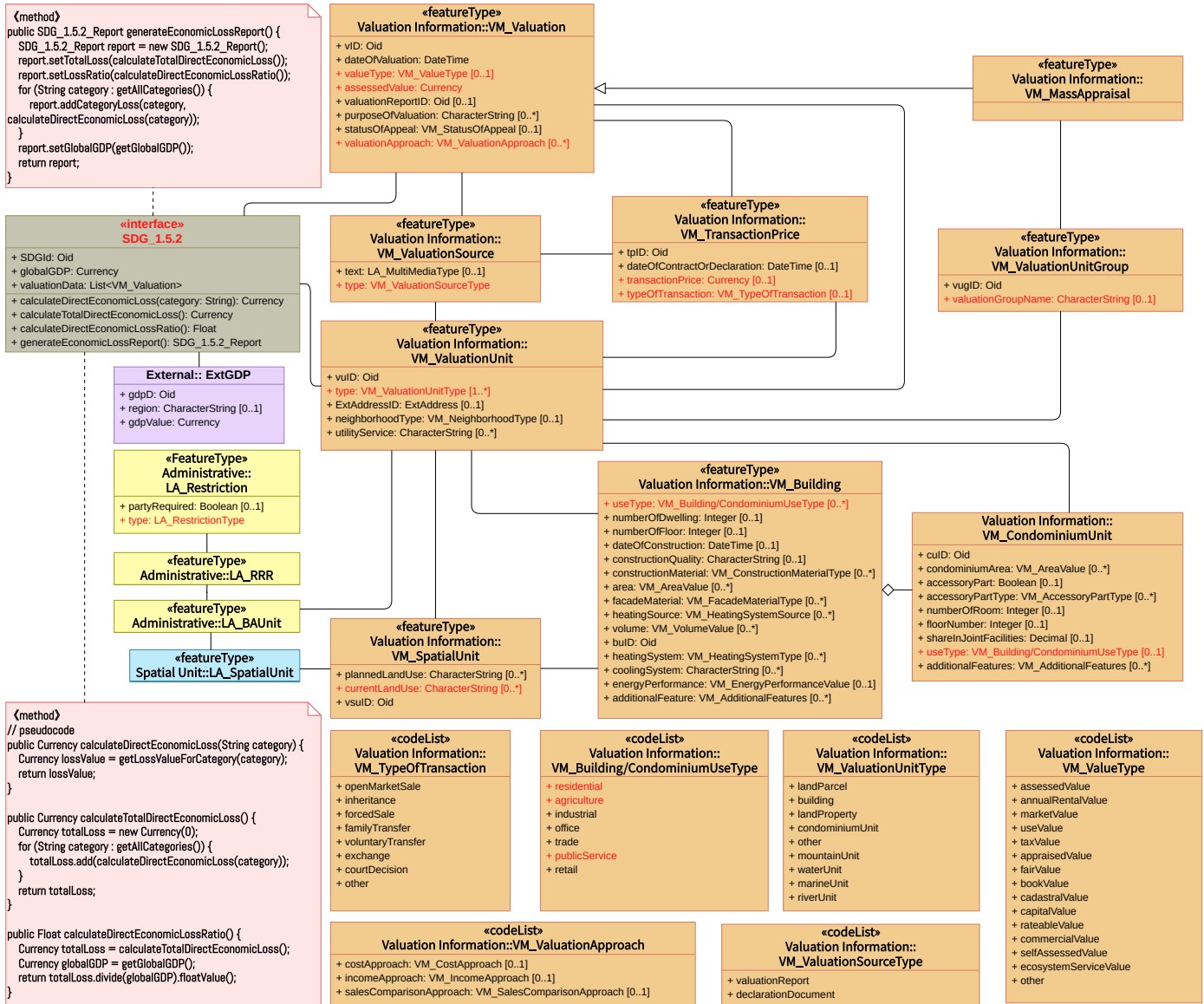

**Figure 8.** Modeling of SDG Indicator 11.5.2 calculation in a UML class diagram.

## 4. Discussion

The detailed examination of specific case studies illuminates the tangible contributions of the LADM to realizing SDG objectives. By integrating the LADM's components with various SDG indicators, the capacity to ensure the security of land rights, promote gender equality in agricultural land rights, manage marine areas for biodiversity conservation and assess economic losses in disaster-prone areas is demonstrated. These practical cases underscore not only the immediate benefits of the LADM but also its potential for global applications, offering mutual benefits within the SDG framework. In detail:

1. **Land Registration and SDG 1.4.2**: The first case study focuses on the integration of the LADM's land registration component with SDG Indicator 1.4.2, which measures the proportion of the adult population with secure rights to land. By mapping legal documentation of ownership and perceived security of rights to key elements of the LADM (such as LA_Party, LA_RRR, LA_Source), this study reveals the framework's capability in ensuring the security of land rights.

2. **Agricultural Land Rights and SDG 5.a.1**: The second case examines agricultural land rights (SDG Indicator 5.a.1) from a gender equality perspective, exploring land rights issues. The land registration and rights components of the LADM are utilized to analyze the proportion of the agricultural population, particularly women, who have secure rights to agricultural land, highlighting the LADM's application in promoting gender equality and land rights.

3. **Marine Area Management and SDG 14.5.1**: The third case turns its attention to the LADM's marine area management and its combination with SDG Indicator 14.5.1 (the coverage of protected marine areas). The application of the LADM in defining and managing protected areas and marine zones demonstrates how effective management of marine protected areas can support biodiversity conservation goals.

4. **Land and Property Valuation and SDGs 11.5.2**: The fourth case explores the relationship between the LADM's land and property valuation component and SDG indicators 11.5.2 (direct economic loss attributed to disasters as a proportion of global GDP). Through estimating and calculating economic losses, this case showcases the potential application of the LADM in disaster risk assessment and management.

Furthermore, the insights derived from these case studies have significant policy implications. Policymakers can strategically incorporate the LADM into land management frameworks, streamlining processes and enhancing the overall efficiency at both local and national levels. The focus on gender equality in agricultural land rights suggests policies tailored to address gender disparities, emphasizing the LADM's role in this critical aspect. Marine area management policies, informed by the LADM, can contribute to biodiversity conservation goals, showcasing the potential for environmental sustainability. Additionally, integrating the LADM into disaster risk assessment and management policies provides a robust foundation for decision-making processes related to disaster risk. These recommendations offer a practical roadmap for leveraging the LADM to achieve sustainable development goals, providing actionable strategies for policymakers and organizations to enhance their land management policies.

This research also holds substantial implications for decision-making processes at various levels, spanning from local to international spheres. The insights derived from the case studies provide actionable guidance for policymakers. At the local level, municipal and regional authorities can optimize land management practices by integrating the LADM into local policies, positively impacting communities. Nationally, governments can formulate comprehensive land administration policies aligned with SDGs. Internationally, this research contributes to discussions within ISO TC211 and the Land Administration Domain Working Group of the Open Geospatial Consortium (LandAdmin DWG), informing the development of specific LADM revisions. It is also helps to support the "standards" in the nine pathways proposed by FELA (Framework for Effective Land Administration).

However, limitations exist in this methodology that need further exploration. The initial step of the Four-Step Method, Step 1: Keyword Extraction and Preliminary Filtering, currently relies heavily on manual literature reviews. Future iterations may benefit from integrating advanced semantic network technologies and ontology-construction methods. Despite the potential for these technologies to streamline the process, they also introduce the possibility of errors that need careful consideration and mitigation. Another limitation relates to the adaptability of this method. The current approach is designed for indicators with explicit calculation formulas. Challenges emerge when dealing with indicators lacking clear formulas or those of a complex nature, as seen in the case of Indicator 1.4.2(b), which assesses the perceived security of land tenure rights. Effectively addressing these nuances

requires a more detailed and nuanced approach, potentially prompting an expansion of the methodology's scope.

## 5. Conclusions

This study explores the substantial contribution of the LADM to realizing SDG objectives by screening LADM-related indicators and employing a systematic "Four-Step Method" to formalize four case studies. The findings highlight the LADM's significant potential for improving land use planning, management efficiency, and sustainable land administration. The individual case studies affirm the LADM's pivotal role in advancing crucial areas such as land registration, rights assertion, and information management, aligning with the targets of land-related SDGs.

In addition to these contributions, this paper proposes enhancements to the ongoing revision of the multi-part LADM standard, especially Part 6. These include novel procedures for calculating indicators, the integration of blueprints for external classes addressing additional information needs, and the design of interface classes for displaying indicator values specific to countries and reporting years. This study underscores the profound potential of the LADM in supporting SDG achievement and provides a systematic approach for its integration into sustainable land administration practices. The presented "Four-Step Method" for formalizing SDG indicators is versatile, suggesting compatibility with other (ISO) standards. A more formal indicator foundation is advocated to eliminate ambiguities, enhance efficiency in computation and yield more accurate values, truly reflecting SDG realization.

In future research, it should be emphasized that although the theoretical research in this paper provides a solid foundation for the integration of the LADM and SDGs, there are still insufficient case studies in practical applications. Therefore, future work should focus on integrating this theoretical framework with specific practice cases. Specifically, future research should:

1.  **Expand on Implementation Studies**: The theoretical method developed in this research should be applied to actual data from different countries and regions. This will help in assessing the LADM's applicability and effectiveness in diverse legal and cultural settings.
2.  **Identify and Address Gaps**: By integrating real-world data, future work can bridge the gaps between current land management practices and the ideal scenarios envisaged by LADM and SDG integration. Identifying these gaps is crucial for developing targeted strategies to address them.
3.  **Improve Policies and Systems**: Use the insights gained from practical applications of the LADM in supporting SDGs to provide concrete recommendations for governments and policymakers. This will aid in improving land management systems and policies to better align with sustainable development goals.
4.  **Conduct Formal Standardization**: The obtained results will be exploited by the editorial team for LADM revisions in order to discuss them within ISO TC211 and/or the LandAdmin DWG. Valuable insights can be used in Part 2: Land Registration, Part 3: Marine Georegulation, and Part 4: Valuation Information, while the calculation of relevant indicators could be part of discussions regarding the development of Part 6: Implementations.

Various SDGs and indicators lie within the scope of the LADM; however, the presented Four-Step Method to formalize the SDG indicators is generic and it is expected to work well in combination with other (ISO) standards, like ISO19144-2 [34] Land Cover Meta Language (LCML) and ISO19107 [35] Geographic information—Spatial schema. Future work related to a formalization similar to the one presented in this paper of all SDG indicators will result in more accurate indicator values and calculations, facilitating true SDG realization.

**Author Contributions:** Conceptualization, P.V.O., E.K. and M.C.; methodology, P.V.O., E.K. and M.C.; formal analysis, P.V.O., E.K., M.C., P.D. and C.L.; investigation, P.V.O., E.K. and M.C.; writ-

ing—original draft preparation, M.C. and E.K.; writing—review and editing, P.V.O., E.K., M.C., C.L. and P.D.; supervision, P.V.O.; project administration, P.V.O., E.K. and M.C. All authors have read and agreed to the published version of the manuscript.

**Funding:** This research received no external funding.

**Data Availability Statement:** No data was used for the research described in the article.

**Conflicts of Interest:** The authors declare no conflicts of interest.

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
