# Peer review of "Bridging Sustainable Development Goals and Land Administration: The Role of the ISO 19152 Land Administration Domain Model in SDG Indicator Formalizationâ€"

_land, doi:10.3390/land13040491_

Round 1
Reviewer 1 Report
Comments and Suggestions for Authors
I am very impressed by the quality of this paper. The authors tackle the difficult problem of relating land management, formalised in the ISO standard, to the indicators of the SDGs, which, according to many researchers, are undefinable. What deserves special attention is the simplicity of the language, which makes the text understandable to a wide range of recipients, not only academics, but also local and government authorities and practitioners.
The authors have demonstrated enormous theoretical and practical knowledge, which is evident in the literature review, the formulation of the research problem, the methodological approach to solving it, and documenting results.
Author Response
Please see the attachment "Response to Reviewer 1 Comments" at page 1.

Reviewer 2 Report
Comments and Suggestions for Authors
The manuscript contains useful ideas and is worthwhile for the Land journal. However, some aspects need to be corrected to improve comprehensibility and appropriateness.
1- The title "Bridging Sustainable Development Goals and Land Administration: The Role of ISO 19152 LADM in SDGs Indicator Formalization" effectively communicates the main issues. However, it could be more concise without losing meaning. Consider adding specific SDGs addressed in the study to give readers a clearer idea of the approach.
2- To better engage readers, highlight in the abstract the novel contributions and results of the study, such as specific improvements to LADM and the impact on the calculation of SDG indicators.
3-The manuscrpit should engage the reader by clearly stating the problem addressed, the methods used, and the outcomes achieved.
4-Connect the study's findings to broader implications for SDG implementation, policy-making, and global sustainability efforts.
5-The literature review in this manuscript provides a fundamental understanding of the research context, although it could benefit from greater depth and breadth. While key concepts and frameworks have been introduced, there is scope to broaden the discussions by including more recent and diverse sources. For example, recent studies on research trends in land tenure. E.g.
(2023). Taking stock of global land indicators: A comparative analysis of approaches for a globally consistent land tenure security measure. Land Use Policy, 124, 106376.
(2023). Worldwide research trends on land tenure. Land Use Policy, 131, 106727.
(2023). Urbanization and Land Use Planning for Achieving the Sustainable Development Goals (SDGs): A Case Study of Greece. Urban Science, 7(2), 43.
6- Highlight the potential real-world impact of the research. Discuss how the proposed framework can be implemented by land administrators, policymakers, or international organizations.
7- Discuss potential challenges or limitations faced during the case studies and how these were addressed or could be addressed in future applications.
8- Provide more insights into how the proposed methodology aligns with existing ISO standards, and how it could complement or enhance them.
9-Elaborate on the concrete policy recommendations that can be derived from the study. How can governments or organizations use the findings to improve their land management policies?
10- Consider discussing the implications of the research on decision-making processes at various levels, from local to international.
11- Provide more details on how the proposed enhancements could be incorporated into specific parts of the LADM standard, including Part 2: Land Registration, Part 3: Marine Georegulation, and Part 4: Valuation Information.
12- Emphasize the importance of accuracy in indicator values and the role this plays in showcasing true SDG realization.
Comments on the Quality of English LanguageIn general, the English of the manuscript is clear and understandable.
Author Response
Please see the attachment "Response to Reviewer 2 Comments" on pages 2 to 5

Reviewer 3 Report
Comments and Suggestions for Authors
The objective of this paper was to establish links between LADM and SDGs by introducing the "4 Step Method" to formalize the SDG indicators within the LADM framework. This is a great use case for future implementation of LADM as the realization of SDGs is a global objective. Therefore, the narrative of this research and the developed methodology should be fitted in the context of Part 6: Implementation of LADM II. This is supported by the fact that the presented case studies utilize classes and data from other LADM packages. In this way Parts 1-5 could be connected by this use case of LADM in one of the future implementations. Notably, Part 6 is only mentioned to exist in the Introduction, and briefly in Conclusion as an objective for future research.
Overall, the quality of the paper is good and provides research novelty regarding the LADM.
Some of the suggestions to further improve the quality of the paper can be found below:
· Line 33 and 34 – Here the authors mention that LADM aims to support the computations of relevant SDG indicators in future parts of its development. From this paper it might be concluded that by using the LADM based land administration system (LAS) it could be possible to formalize and compute certain SDGs. This seems like this research might contribute to future part of the LADM which is currently in development, namely Part 6: Implementation. If LADM was to offer examples and methods for formalizing and computing SDGs using LAS data, I believe it could be introduced in Part 6, as this might prompt for future use cases of implementing LADM in different jurisdiction areas. However, reading thorough the paper the Part 6 of LADM development is mentioned very briefly.
· Lines 39 and 49 – In this part of the Introduction the notion of SDGs formalized in this paper should be mentioned clearly both with the numbers and names as it was in the abstract of the paper, e.g. securing land rights (1.4.2), agricultural land rights (5.a.1) etc. Using only numbers might require from readers to check each of the indicators outside of the paper resulting with difficulties in reading the paper.
· Lines 43 and 44. The authors state that the current process of calculating SDG indicators is complex, labor-intensive and error-prone. This should be explained in more detail in the Introduction as it might emphasize the need for the developed 4 Step Method.
· Lines 78 and 79 – In these lines it is stated that the second edition is under development, comprising six parts, while the provided Figure 2 represents the development of 5 parts. It should be noted what is Part 6 about, and acknowledged that it is currently in early stages of development.
· Lines 118-122, Keyword extraction seems like a critical point of the developed methodology, however it is not stated how it was conducted. Was the keyword extraction conducted manually, reading through the SDGs or automatically via deep learning or something else? Furthermore, highlighting the importance of unified and standardized terminology is crucial for such procedure. This could be noted either here or in conclusion as possible obstacle in analyzing the SDGs.
· Lines 199 and 200. The last sentence of paragraph should be rephrased as it is not clear what it says. I guess it was meant to explain that some of the data required for calculating the indicators could not be provided by LADM and should be used from other sources.
· Lines 333-342. In these lines authors explain colors used in the diagrams for Figure 5, however, some of the classes you mention here are not used in Figure 5, e.g. orange for Valuation Information Package. This kind of information should be either mentioned in methodology (e.g. section 3.4 Create UML) or before every Figure depicting the case studies.
· Line 632, authors mention five case studies while there are four in the paper. Four cases are listed below this sentence as well.
· In line 688, authors mention that the developed methodology is expected to work well with other (ISO) standards. This might be relevant for future research as standards alignment is expected and, in this way, it might be possible to address links of LADM with other standards and finally with SDGs, or use SDGs to link LADM and other standards. Identifying some of the possible standards to link with LADM and SDGs might be helpful.
· References list should be completed with all data, e.g. papers from proceedings should include place and date of the conference.
Author Response
Please see the attachment " Response to Reviewer 3 Comments" from page 6 to 10.

Round 2
Reviewer 2 Report
Comments and Suggestions for Authors
The authors have satisfactorily addressed my suggestions.
Author Response
Thank you very much for your valuable comments and suggestions during the review process. We are truly grateful for your insights and pleased to learn that you feel we have satisfactorily addressed all of your previous suggestions.